# An Exploratory Study of Crime: Examining Lived Experiences of Crime through Socioeconomic, Demographic, and Physical Characteristics

Christopher Chimaobi Onyeneke * and Aly H. Karam *

School of Architecture and Planning, University of the Witwatersrand, Johannesburg 2000, South Africa
* Correspondence: christopher.onyeneke1@students.wits.ac.za (C.C.O.); aly.karam@wits.ac.za (A.H.K.)

**Abstract:** Debates in urban social research indicate that one of the most significant problems facing cities of the global south is the dilemma of crime. This study explores the research question: what is the level of influence of economic deprivation and residential mobility weights on criminal activities within vulnerable neighborhoods in urban centres? This research focuses on the urban social structural theory of social disorganization, to investigate how socioeconomic, demographic, and physical characteristics affect criminal behaviour. The study adopted a qualitative research approach that is cross-sectional. With the use of volunteer self-selection sampling techniques, in-depth interviews were conducted with heads of households via a semi-structured interview guide. The data were analyzed using thematic analysis. The study found that sociological factors such as economic deprivation and socioeconomic inequality lead to the gradual integration of delinquency in cities. The findings of this research build on the existing theory of social disorganization. To ensure safety, residents' economic status must be standardized by supporting the provision of more regulated informal sector opportunities and activities for active engagement in supervising and controlling youth behaviour.

**Keywords:** social disorganization; socioeconomic inequality; vulnerability; unemployment





## 1. Introduction

The word 'crime' originates from the Latin word 'crimen' meaning 'charge' or 'offence.' Crime is a social fact, a dangerous phenomenon in any city, region, or nation [1]. According to [2], violence is defined as "The intentional use of physical force or power, threatened or actual, against oneself, another person, or against a group or community, that either result in or has a high likelihood of resulting in injury, death, psychological harm, maldevelopment or deprivation" [3]. Violent or personal crimes within cities and urban centres include abduction, affray, burglary, hooliganism, kidnapping, riot, robbery, looting, lynching, manslaughter, pickpocketing, mugging, hit and run, murder, rape, shoplifting, homicide, smuggling, theft, assassination, assault, trespassing, hijacking, and vandalism. Under the United States Bureau of Justice Statistics, Perkins reported that people aged 14–25 are at the peak age for criminal activity [4]. Most of these crimes occur among youths and juveniles. When these crimes are committed, people ask what is wrong with society and who is to blame? Minor issues in the community or society, such as hooliganism, vandalism, and pickpocketing, gradually evolve into something more serious and substantial such as mugging, robbery, theft, abduction, and even murder. As financial stakes rise, the link between drug abuse and violence gradually becomes part of a subculture within the community [5].

Recent developments in urban social research such as [6–8] have led to a renewed interest in safety and security. These are indeed the core components in creating safe residential complexes and cities. In the past two decades, people have built fortresses to make their residences safe and secure. Researchers such as [9–11] have shown that, despite significant changes in urban morphology and growth, the issue of creating a secure and safe

living environment for inhabitants has been at the forefront of discussions on urbanization and urban management. One of the hypotheses of crime is that it is motivated by economic deprivation [12]. Residential segregation in urban areas is an essential mechanism for the causes of crime [13]. Moreover, it is a central part of the socio-physical recovery of the excluded urban sectors. The main factors of crime in urban centers are homelessness and deprivation, poverty and unemployment, youth marginalization, the disintegration of culture, poor urban planning, and neglect of public spaces [14]. Crime results from the perception–choice process and the criminogenic condition of the residents' living environment. Social disadvantages are not the causes of crime but are critical factors which lead to criminogenic exposure and crime propensity in residents [15,16].

In support of the debate on economic deprivation and crime, a recent study [17], to fill the gap in the existing literature, focused on medium-size Latin American cities to examine the influences of the built environment on street walkability and the perception of crime. The result reveals that security within residential neighborhoods is highly correlated with neighborhood quality, and this situation of insecurity has the most significant impact on walkability and the perception of crime within the city. In the past three decades, many researchers have sought to determine the root causes of crime from a long list in order to devise means to tackle it. A study of American cities highlighted that since the 1980s inner-city crimes have become a national issue and a public concern [18]. These rates tend to be higher at the city core and have shown a high rise, particularly in cities with an estimated population of more than two hundred and fifty thousand people. Youths have been the major victims of these crimes and even in severe situations this has led to deaths. The violent crime statistics show that the number of young people between the ages of 10 and 16 years, whose annual household income is below $10,000 and have ever been assaulted, is higher in major cities, especially in the global south [19]. This clearly shows that crime is related to socio-economic class, geographical location, and population density.

The streets of our towns and cities have been places where danger is rife, and death is commonplace since the birth of urban civilization some centuries ago [20]. Natural, human-induced disasters and mass migration of rural dwellers to urban centres have significantly increased the factors involved in crime [21]. A considerable amount of literature has been published on the perception of crime in cities. Several studies have argued about the different issues surrounding the causes of crime in cities. Moreover, this question has continued to gain attention, and this research has tended to fill the gap in the existing literature on the influence of economic deprivation and residential mobility on criminal activities within vulnerable neighborhoods in the urban center, with more citizens' involvement, and while examining the socio-economic, demographic and built environmental characteristics on lived experiences of residents.

This research investigates the determining factors of crime in cities of the global south using economic deprivation and residential mobility data, particularly in the Borokiri neighborhood of Port Harcourt city. It seeks to investigate the extent to which economic deprivation and residential mobility influence crime in this neighborhood. Port Harcourt city is ranked number one in violent crime among cities in Nigeria, with high crime rates including vandalism and property crime [22]. Crime is a significant phenomenon across the cities of the world, including Borokiri, and has the propensity to affect the general well-being of urban residents' quality of life. Most residents in urban areas, especially those in Borokiri and Port Harcourt, live in constant fear and apprehension because of the fear of crime. Intending residents who are aware of such high crime rates choose to decline the offer in favor of safer neighborhoods they can afford.

Furthermore, in carrying out this research, the data collected for this study include socioeconomic, demographic, and environmental characteristics of urban residents, particularly in the Borokiri area, and the role each play in vulnerability and crime. This study adopts a qualitative research approach that is cross-sectional, using in-depth interview and observation techniques. The volunteer self-selection was adopted to select respondents across the different groups of stakeholders including the king of the community, clan heads,

public security agencies (police), religious groups, principals of secondary schools, local government representatives, youths, and market women. 62 interviews were conducted with those who have lived for more than ten years in the neighborhood to obtain a holistic account of the lived experiences of residents.

In addition, from interviews with selected respondents, the findings of this study show that socioeconomic forces at work in the Borokiri neighborhood are factors significantly contributing to the area's high crime rate. The level of susceptibility in the neighborhood is influenced by social characteristics such as education, economic position, social activities, and peer groups. This research finds that social disadvantage and economic deprivation/marginalization are the root causes of crime, especially in the Borokiri context, and that heterogeneity has no relationship with crime. Moreover, it is often stated that crime is correlated with age and gender, in the Borokiri neighborhood, crime has no relationship with age, and neither does it with gender. More so, this research finds that poor infrastructure puts the neighborhood in a disadvantageous position. The economic depression which manifests in physical characteristics in the case of Borokiri in Port-Harcourt poses challenges in managing criminal tendencies which in turn makes room for delinquency and crime, which gradually becomes a sub-culture.

There are several arguments in the social sciences about issues of economic deprivation and how they affect crime but have failed to look at the fact that the social structure of society is experiencing a shift from the norm (social). This change is influenced by unemployment, economic deprivation, inequality, and the concept of the value of education. In order to tackle the challenges of crime in our society, the impact of unemployment, economic deprivation, and inequality must earnestly be placed as a top priority to achieve the goal of national development. As a result, in the subsequent sections, this paper goes on to review relevant literature and works regarding crime in cities, analyze the data collected through in-depth semi-structured interviews with the residents of the Borokiri neighborhood of Port Harcourt city, and suggest possible measures and strategies through which crime can be controlled and further occurrences prevented.

## 2. Literature Review

Crime—as a social phenomenon—is inspired by several forces which can be traced to societal constructs. It is often a function of unaddressed social determinants present in society. When these factors are not identified or not properly treated, they result in significant increases in societal crime rates. This leaves the society in disorder and chaos and unsafe for residents. Moreover, social forces operating in urban areas are critical factors responsible for crime in cities. The social disorganization theory which suggests that crime is a function of the disintegration of social control in metropolitan areas evolves from the work of Chicago urban sociologists in 1942, led by Shaw and Mckay. These social forces include economic status, education, religion, belief system, social media, and peer groups. Overwhelming scientific evidence has proven that these intrinsic urban factors motivate crime [23]. However, [24] argues that social forces can most efficiently fight against crime if they are properly harnessed, i.e., if resources are concentrated on preventing chaos, social nuisance, and insignificant offences such as vandalism, drinking in public, loitering, rowdiness, and disorderly behaviour, as well as improving dilapidated physical structures in urban areas [25].

The city has three distinct areas: areas for the affluent or wealthy class, the middle or transition zone, and areas suffering from depressed infrastructural facilities, poverty, and disintegration [26]. This distinction is regardless of religion, race, and ethnicity [27]. Social forces influence the behaviour and actions of people living in these areas within an urban setting, which leads to the evolution of regions dominated by crime [28]. The theory of social disorganization holds that low-income residential neighborhoods plagued with the breakdown of social control in families, schools, and religious bodies are the focal point for crime in urban centres [29]. As seen in Figure 1, the theory emphasizes that the urban social environment (sociological factors) significantly influences crime. A disorganized

urban setting is one in which institutions of social control is weakened and broken and its ability to carry out its normal functions is hampered [30]. Thus, gradually the city becomes socially chaotic, hence difficult to control and to protect residents against crime and the criminal behaviour of offenders.

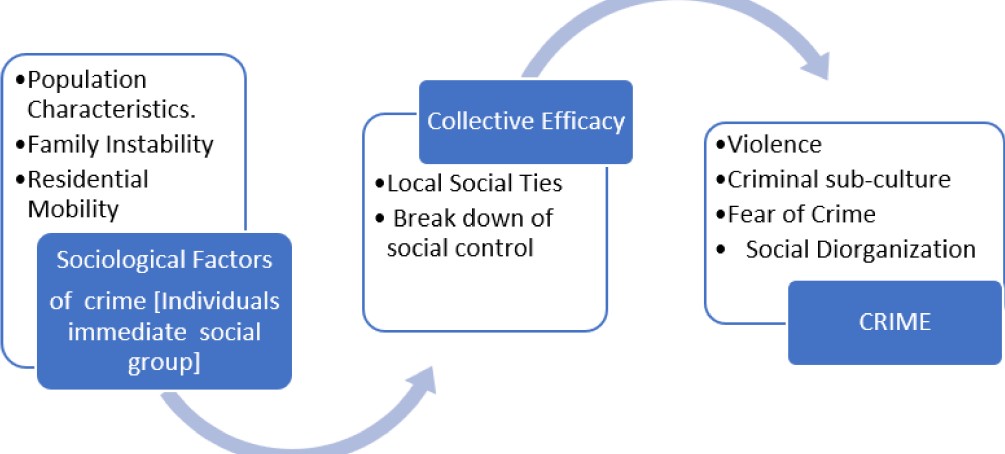

**Figure 1.** Social Disorganization Theory.

There is a retrogression in the social life of residents in sections of cities facing challenges in the provision of essential services such as decent housing, educational facilities, and quality health care for residents. Consequently, a delinquent/criminogenic sub-culture gradually develops [31]. Residents in these crime-ridden areas of the city tend to flee when they no longer feel comfortable with the turn of events in the neighborhood and new residents with no knowledge of crime in the neighborhood replace them [32]. Gradually residents become uninterested in community matters. The neighborhood they occupy becomes destabilized as inequality continues to grow in the city, and more sections of the city fall into the fray [33]. Crime leads to constant population turnover, which weakens communications, blocks the actualization of goals, and slows down development. The residents' nonchalant attitudes lead to streets being littered and untidy, and the urban fabric deteriorate. These neighborhoods are gradually rezoned out of necessity to mixed-use (residential and commercial properties next to each other) and are difficult to control by the usual social agencies [26].

The issue of mixed land uses makes the urban space difficult to manage and crime-prone, and crime affects the everyday life of urban residents, the quality of urban space, and its socio-economic and cultural prospects. The consensus report from studies [34–36] shows that the overall impact of insecurity limits socio-economic development in various ways. Crime and insecurity in an urban setting create social dislocation, promote population displacement, increase social tensions, and encourage hostility. This, in turn, deepens poverty, discourages local and foreign investment, halts business operations, increases the cost of doing business, and reduces property value. Members of society become vulnerable and create an atmosphere of mistrust, fear, anxiety, and frenzy, etc. [37].

The rise in crime affects the economic situation. Economically, urbanization is positive as it facilitates development and thus promotes urban growth and the provision of services [38]. However, according to urbanization indices, crime is promoted in urban centres where the population is heterogeneous and where there is limited security for citizens due to the high population [39]. As shown in several observational reports, crime levels are higher in larger metropolises and cities. While several crimes are relative to urbanization, however, many other determinants directly affect the increasing crime trend in urban areas. These determinants include unemployment, inequality and income discrepancy disparity. However, these relative determinants are often linked to urbanization or its consequences, showing that the root causes remain in the urbanization process [9]. Before now, many

empirical studies [9–11] have shown an urbanization–crime nexus, and suggest that both terms are correlated The universality of that correlation has, however, not been established.

In the same vein, Heinemann [40] argues that the severity of crime issues varies significantly by region. In the Andean region across South American countries, the violent crime rate seems to be 51.9 per 100,000 against 6.2 in the southern part of these countries. Since the 1970s, overall crime rates have been on a steady increase. Urban areas, especially major cities such as Rio de Janeiro, Lima, Mexico City, and Caracas, seem to be the most vulnerable to violent crime. Crime is discussed from three perspectives: social effects, root causes of conflicts, and empirical crime determinants. There is a general agreement that injustice is the major cause of the high crime rate in Latin America. Apart from income disparity, major risk factors include low rates of schooling, low social capital, homelessness, urbanization, and an inadequate criminal justice system. Therefore, prevention initiatives are essential to combat the rising issues of crime.

A longitudinal study [41] investigated the relationship between official crime rates in census tracts and resident perceptions of crime. Drawing from the data set of the Housing Survey samples in America for over 25 years (1976–1999) along with the census tract data on crime rate from the selected cities, the study reveals that the perception of crime among residents is strongly correlated to violent crime. The study also reveals that robbery and assault are the most significant violent crimes that substantially affect residents' perceptions. Burglary had a more substantial effect during the 1970s on the perception of crime, but as the years have gone this has declined, weakening its impact. Racial and ethnic composition shows very little significance regarding the perception of crime. Research has consistently shown that crime stems from social, economic, political, psychological, physical, and geographical factors. Just as crime is committed in certain geographical locations, it is worth discussing the relationship between crime and neighborhood socioeconomic and demographic characteristics. A very significant study [10] found that cities' growth and rising development attract people struggling to survive off the land, especially in the movement from rural areas to urban centres, thus causing overpopulation leading to poverty, hunger, violence, and criminality. This is inevitably associated with cities where inhabitants are victimized on urban streets [42].

On the contrary, a qualitative study engaging the chain referral (snowballing) method, conducted on Mexican migrants to the United States, investigated the perception of crime and the extent of crime incorporated as a way of life, according to their experiences. The perception that migrants are responsible for crimes such as theft is explored in comparison to other crime-related issues including risks, vulnerabilities, offending, and victimization. The result shows a low-level relationship between migrants and crime-related activities [43].

As a result, based on the assumptions of the broken windows theory, and from the foregoing, this thesis argues that forces can most efficiently combat crime by focusing their efforts on targeting disorder, social nuisance, and minor crimes such as drinking in public, loitering, rowdiness, disorderly behaviour, and vandalism, as well as combating dilapidated physical structures in urban areas. Disorderly attitudes have been shown to raise crime apprehension, reduce social resilience and contribute to crime. However, the perception of crime was not (as expected) substantially linked to a particular group/class (e.g., youth) in the society. The direct impact of perceived societal dysfunction on expectations of crime was the most decisive influence in any model [44].

Further research on public safety shows that there has been an increase in concern for public security among politicians and public opinion experts, seen in citizen's safety requests from field surveys and datasets from the police consisting of filed reports from citizens in Lombardy, Italy. This method of investigation is unique, and peculiar to the region, aiming to arrive at more insight into how citizens view urban security issues and seek counsel/assistance within their neighborhood. The result shows a need to broaden the concept of urban security towards newer frontiers, particularly regarding quality of life, economic, and social dimensions. The perception of insecurity and crime in Lombardy is

connected to a paradigm shift in the city's urbanistic, architectural, and social morphology, upholding the broken window theory of what causes crime [45].

The causes of crime have been the subject of intense debate, yet this issue has not been studied rigorously and thoroughly in the literature. The causes of crime are yet to be resolved, and this is a legitimate subject of debate [46]. Above, we have shown varied contradictory ideas on the root causes of crime, spanning across the four major urban social structural theories. Suburbanization highlights many issues, indirectly causing social and economic dislocation and isolating inner-city neighborhoods [47]. The argument is that poverty is not the root cause of property crime, though an opportunity is presented [48].

Recent research emphasizes property crime rates in cities that focus on intentional and deliberate attempts to cause property loss [49]. The location and hotspots for crime stress social and environmental roles in establishing causative theory, particularly regarding the interaction between people and public spaces, creating a link between offences, offence inducement and offence tendency [37]. The specific reason for looking at crime causes from this point of view is that it is assumed that the major cause of crime is external. After all, social forces have been argued to affect an individual's engagement and opportunity for committing a crime.

Major concepts used to examine the social disorganization theory include population characteristics, residential mobility, diminishing social ties, breakdown of social control, and criminal sub-culture. These concepts could be examined through family background, employment status, and occupation [50]. Socioeconomic inequality, school dropout rate, and educational qualification are key variables in examining population characteristics concerning crime in cities [51]. In addition, age and gender are very significant indicators/variables in assessing the influence of population characteristics on crime [52].

The population in affected areas tends to exhibit frequent mobility. Residential mobility is another fundamental concept used in the literature to describe the theory of social disorganization. Moreover, racial and ethnic composition and socioeconomic inequality are vital variables used to describe this concept in social disorganization theory [53]. The residential mobility rate in an area and poor interpersonal relationships are significant variables in crime [54]. Proximity to facilities, availability of infrastructure and services, and physical environmental deterioration are critical [55]. The criminal sub-culture explored in the literature is examined through the number or percentage of people who practice deviancy, societal acceptance, and public compliance with illicit acts [56,57].

*Crime in Nigeria*

Crime rates across the globe are on the increase, with most events occurring in the global south. Moreover, crime in the global north cannot be compared to crime in the global south, where the problem is more exacerbated, and crime control strategies seem ineffective. The increase in criminal events happening in the global south have resulted in several cities being plagued with significant rises in crime rates. Amongst the nations significantly affected by rising crime rates is Nigeria, where several violent crimes including religious violence, insurgency, youth restiveness, communal clashes, and several other social vices have torn apart the peace of Nigeria's inhabitants, hence the insecurity in the country. Nigeria is now at the forefront of global statistics on crime [58]. In the last decade, Nigeria has recorded high incidents of violent and non-violent crimes. Nigeria ranks as the tenth most dangerous country globally (as seen in Table 1 below) with two local cities listed among the 20 most dangerous [59,60]. Globally, Nigeria is one of three African nations ranked in the top 20 most dangerous countries, behind South Africa and Namibia, and ranks ahead of war zone nations like Afghanistan and Syria [61].

**Table 1.** Crime and Safety Index Ranking of Countries (Data source: [62] and authors' own calculation).

| # | Country | Crime Index | Safety Index |
|---|---|---|---|
| 1 | Venezuela | 82.38 | 17.62 |
| 2 | Papua New Guinea | 79.49 | 20.51 |
| 3 | Honduras | 79.14 | 20.86 |
| 4 | South Africa | 76.08 | 23.92 |
| 5 | Trinidad and Tobago | 74.20 | 25.80 |
| 6 | Brazil | 70.28 | 29.72 |
| 7 | El Salvador | 68.45 | 31.55 |
| 8 | Namibia | 68.10 | 31.90 |
| 9 | Jamaica | 66.47 | 33.53 |
| 10 | Nigeria | 66.14 | 33.86 |
| 11 | Afghanistan | 66.03 | 32.04 |
| 12 | Iraq | 65.23 | 33.02 |

This result emanated from a perception index of the fear of crime, using a critical variable for assessing violent and property crime within cities. However, this is a perception index, and the actual crime statistics are not considered, but rather the level of safety and security which citizens feel in their cities. The safety index emphasizes how safe citizens feel walking alone during the day and at night within the city or specific areas of the city.

In recent times armed robberies are recurring forms of crime in Nigeria, contributing to about 50% of an overall 8516 crime-related deaths between June 2006 and September 2015. As shown in Table 2, cities such as Lagos, Port-Harcourt, Ibadan, Makurdi, and Onitsha were reported as the most affected [59]. Crime fatalities are more common in the Southern part of Nigeria than in the North, terrorism and religious clashes being perhaps more common. However, the government has shown an inability to address most of the sources of crime, which explains why it has become challenging to nip crime in the bud [63].

**Table 2.** Crime rates comparison between major cities in Nigeria. (Data source: [62] and authors' own calculation).

| # | Country | Crime Index | Safety Index |
|---|---|---|---|
| 1 | Port-Harcourt | 75.59 | 24.41 |
| 2 | Jos | 73.45 | 26.55 |
| 3 | Benin | 71.51 | 28.49 |
| 4 | Lagos | 64.99 | 35.01 |
| 5 | Abuja | 63.45 | 36.55 |
| 6 | Kano | 61.23 | 40.21 |
| 7 | Ibadan | 44.50 | 55.50 |

Port-Harcourt is currently the largest city in Rivers State. It lies in the South geopolitical zone and is ranked the fourth largest city in Nigeria. Its geographical location is in the oil-rich coastal region of the Niger delta. Port-Harcourt has an estimated population of 2,876,763 million as of 2019 [64]. The city has a population density of about 7796.10 persons per km$^2$. The population is spread over a total surface area of 369 square kilometers [65].

In light of the crime rate and demographic characteristics presented above, and because not many studies have been carried out in Port-Harcourt, this is the reason why it was chosen for this study. Port-Harcourt city has significant demographic characteristics such as significant age disparity, ethnic and religious diversity, a significant level of education and income disparity, deteriorating urban fabric and a significant crime rate compared to other cities within the region. The high-crime neighborhoods in the metropolitan city of Port Harcourt are Diobu, Rumuola, Borokiri, Ikwerre, Ogbunabali, Abuloma, and D-line. The neighborhood chosen for this study is Borokiri because it is accessible to the researchers,

accessible by transport and research assistants are available. Borokiri is located south of Old GRA in Port-Harcourt city. The neighborhood of Borokiri is located in Old Port-Harcourt, bounded to the east by Okrika Island (the Aboturu creeks), to the west by Ship Builders Road, by Ahoada Street to the north, and by Orubiri oilfield to the south. Its land uses include residential, institutional, commercial, and recreation. The residential district of Borokiri (as shown in Figure 2 above) is selected for this study because of its peculiarity (issues of social change and crime) relevant to the study. Borokiri has a high indigenous population sharing a common social heritage and living with other inhabitants in the urban centres. In addition, the Borokiri area is economically depressed and has a perceived weak social control set-up, making it ideal for the study.

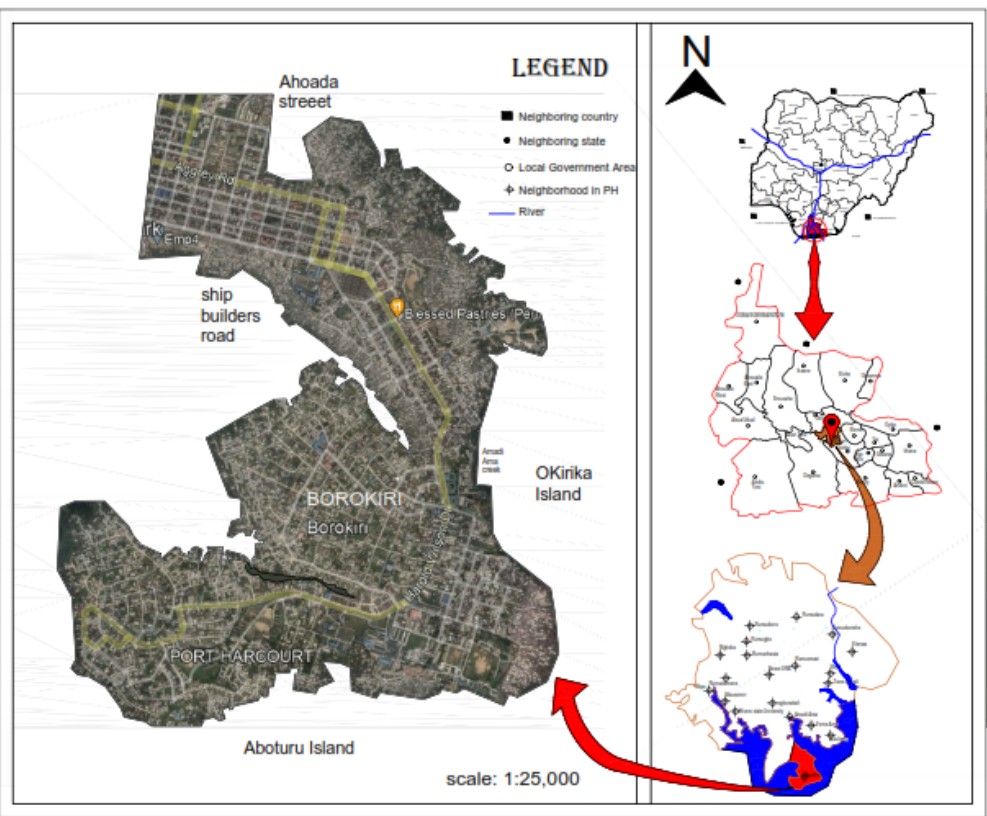

**Figure 2.** Map of Borokiri, the study area.

## 3. Methodology and Data

The purpose of this research is to examine from the experiences of residents the significance of economic deprivation and issues related to neighborhood characteristics on crime. This study explored these impacts from the perspective of the social disorganization theory, and the impact on opportunities to commit crime. The basic data obtained for this study were data from residents' experiences relating to socioeconomic status, economic deprivation, and issues around neighborhood characteristics of the study area. The research adopted the methodological choice of the multi-method qualitative research approach. The method of data collection chosen is an in-depth interview with the use of a semi-structured interview guide.

This research used non-probability sampling techniques of volunteer self-selection sampling. The reason for this selection was that access to respondents was difficult, and the research purpose is exploratory. This approach enabled an in-depth insight into the phenomenon under investigation from the experiences of the local dwellers within the region. Using volunteer self-selection sampling techniques, interviews were conducted with a semi-structured interview guide with 62 adults from the various sectors. The

following are some of the organizations also selected for interviewing: the police, vigilantes, traditional heads of the community, heads of Youth Associations, heads of the market women association, principals of secondary schools, heads of religious organizations, representatives of local government, heads of Trade Union Associations, and captains of industries. The 62 interviews constitute an ideal size for a semi-structured interview program whose target population is heterogeneous [66]. The respondents were all above the age of 30 years and could provide details of lived experience within the community over a long time.

A semi-structured instrument was developed to enable the research respondents to feel free to participate in the research. A checklist was prepared for the observation. The in-depth interview was conducted with 40 adults in the community and was conducted in an informal conversational manner in a quiet environment at an agreed, at their residence or places convenient to the interviewee. A digital recorder was used to capture the discussions/information/data which was then transcribed verbatim to capture every detail. There were also notes taken to ensure accuracy and a precise transcription. For those parts of the interviews that were conducted in Igbo language or pidgin English, a careful translation was made so as not to lose the exact content and original source. Participants' behaviour was recorded using observation guidelines. The study used Scopus and Web of Science search engines for reviews of relevant literature from journal articles and proceedings. Thematic analysis was adopted to analyze the data collected from the qualitative research method using the in-depth interviews conducted. Qualitative data was processed vertically and horizontally at the preliminary stage. The variations and commonalities of the respondent were taken into consideration. Appropriate codes were drawn out of the transcribed interviews, which were later categorized into themes.

This research obtained its ethical approval from the University of the Witwatersrand, Human Research Ethics Committee (Non-Medical) with protocol number H20/04/18, via a rigorous process of attending an ethics course, undergoing the ethics test, being certified, submitting of ethics application, review, issuance of the ethics approval and progress report. The research participants' consent was sought by informed consent form. Research participants were briefed in detail about what the research was about and the implication of participation via a participant information sheet. The anonymity of the individuals involved in the research, who made impersonal contributions to the study, was acknowledged. Moreso, the participants were made to know that they were free to quit the research at any time, without being questioned. Participants were free to discontinue the study, as well as to neglect questions they did not want to answer. These details were captured in the participant information sheet. There were no experiences of distress or discomfort at any point in this process, otherwise the interview would have been stopped, to resume another time.

## 4. Results and Discussions

Port-Harcourt is one of the top five most populous cities in Nigeria with a population of 1,133,019 persons covering a land area of 178 km sq and population density of 6365 persons/sq km. Borokiri in the Old Port-Harcourt town has a very high population of low-income residents of about 95,182 persons according to the National Population Commission of 2021, covering a land mass of 4.8 km sq, i.e., 19,829 persons/sq km. From the findings of this research, the residents live in row buildings of apartment unit. Moreover, this research reveals that most people living in this area are self-employed or have not been previously employed, and very few residents work in artisan and peasant business activities such as restaurants, bars, and pubs. This is emphasized by one of the respondents (ID-00031): "the people who live here are predominantly self-employed, and they try not to engage in illegal activities, but that does not solve the big problem of crime such as cultism and stealing in the neighborhood".

Who commits the crimes in this neighborhood? Only a few residents are civil servants with verified jobs. The youthful population are the largest, as very few elderly people are

seen within the neighborhood. Moreover, school pupils are seen around the neighborhood all day, as school dropout is at an alarming rate. Most of the interview respondents either do not finish secondary school education or are very reluctant about the opportunities that may arise from education. This position was further emphasized by one of the respondents (ID-00021), addressing the opportunity individuals can derived from education: "However, they are self-employed but lack educational background with no idea of creating employment". This suggests that the inability and the reluctance of the youths of Borokiri to engage in productive activities is a function of the lack of educational qualification. The ideas needed for the creation of employment for youths, especially in the Borokiri context, demand quality education. Unfortunately, crime rates in the area continue to rise because the youths do not possess the educational qualifications to provide themselves with lucrative job opportunities, and are reluctant to engage in legal work, but rather resort to crime for their livelihood.

This research found that social forces operating in the Borokiri neighborhood are factors responsible for making the neighborhood crime-ridden. These social factors such as education, economic status, social activities, and peer groups influence the level of vulnerability in the neighborhood. These factors are highly significant in Borokiri, Port-Harcourt, and significantly contribute to the vulnerability of the neighborhood. Several respondents further reacted to this; from one of the vigilantes (ID-00045): "Crime has nothing to do with age nor gender but has a relationship with income; for family size issues, one cannot tell". Another participant (ID-00023) suggests that "It depends on the upbringing of each individual, and it is not easy to change people, but the body of Christ is doing a lot. One thing that strikes an imbalance for crime is poverty, and it happens mainly with our teenagers and youth, especially the male gender. Sometimes, income level can make one commit a crime". According to a member of the law enforcement personnel (ID-00050): "The youth are the perpetrators of this crime because of frustration, peer pressure, and bad friends. However, this area can be safe compared to other areas".

This implies that, while it is stated in most research that crime correlates with age and gender, in the Borokiri neighborhood, crime has no relationship with age, nor does it with gender. This contradicts the social disorganization theory (age) and the general strain theory (gender) as both do not relate to crime but rather, socioeconomic factors (such as income, social support, and community safety) and sociodemographic factors (level of education, employment, and living arrangements) significantly affect the opportunities to commit a crime. In agreement with [23,28] urban social forces motivate crime in cities, especially in urban neighborhoods characterized by gross demographic and socioeconomic challenges and setbacks. As earlier stated, the Borokiri area of Port Harcourt has been highly neglected in terms of facilities and social services compared to some other areas of the city. This supports the ideas of [26,27] that the city has two distinct areas, the areas of the affluent or wealthy class, and that of those suffering from depressed infrastructural facilities, poverty, and disintegration. From the findings of this research, and in agreement with the results of [34–36], in the Borokiri context crime affects the everyday life of residents, neighborhood space quality, and limits the socio-economic development of the neighborhood in various ways.

Several participants residing in the Borokiri neighborhood made contributions, with one of the school principals (ID-00030) stating that "this neighborhood is plagued with poor infrastructure like poor street lighting, water, sanitation, and health care. Companies, industries, and factories are not available, and this has somehow led to the increase in crime. The most common crime within this neighborhood is assault, fighting, and others". Another respondent, a head of the market women association (ID-00025), asserts that "the younger people mostly commit these crimes, and I think one of the major factors that cause crime is poor and inadequate social infrastructure". In the words of another respondent, a leader of a youth organization (ID-00015), "In my understanding crime is committing atrocities, and that occurs everywhere and in this neighborhood. The most occurring crimes are kidnapping, assassination, and stealing. These crimes are due to economic and social

marginalization. This neighborhood has been neglected by the government for a long time now".

Examining crime motivators in the Borokiri neighborhood via the comments of the respondents above, this research finds that poor infrastructure, especially lighting (electricity) puts the neighborhood at a disadvantageous position, inclined to criminogenic probabilities. This supports the theoretical assumptions of the broken windows theory. In the global south, the situation is more complicated as there is visually no access to this kind of physical infrastructure in most crime-ridden neighborhoods. Not that they have been vandalized or show signs of no maintenance, but that they are completely absent. This is because neighborhoods characterized by gross darkness and no access to electricity to light up the dark corners of the area increases the risks of crime perpetration. Crimes including rape, assault, theft, burglary, and even kidnapping could be perpetrated without the knowledge of other residents. Furthermore, in agreement with [29], this research suggests that low-income residential areas including Borokiri are plagued with the breakdown of social control in schools (dropout, vandalism, and insubordination), families (disobedience, fighting, quarrelling and stealing), and religious bodies (anti-social behaviour and activities), the focal point for crime breeding in urban neighborhoods, especially in the Borokiri neighborhood.

In addition, in the global south, the situation is exacerbated as school authorities no longer have control over their pupils. The value of education has been downplayed as the economic situation in the country becomes worse. Pupils in secondary school and now in primary school indulge in anti-social behaviour such as cultism whose foremost aim is rebellion against constituted authority, and vandalism. Moreover, due to the multiple forms of poverty, families have been ravaged with many forms of maladjustment. Parents no longer have control over their children, children no longer respect parents and elders as they do not provide for their daily needs, and wives no longer respect their husbands and vice versa. These children who are uncared for commit all manner of atrocities within their society such as quarrelling, fighting, stealing, and vandalism. Religious and social organizations have compounded the issue as residents now see them as a means of exploitation and swindling than informal institutions of social control. When all these issues continues unabated, the neighborhood gradually becomes socially disorganized and difficult to control and it is difficult to protect residents against crime and criminal behaviour from offenders.

According to a respondent, a member of the law enforcement personnel (ID-00053): "The rate of crime in this neighborhood is usually higher than in any other place in Port-Harcourt". Another participant, also a law enforcement agent (ID-00056), suggests that "crime is a sensitive issue; the common crimes within this neighborhood include pickpocketing, kidnapping, vandalism of crude oil lines, robbery cases, burglary, rape, drug abuse, cultism, etc.; these are happening here". Speaking with another respondent, a community leader (ID-00033), the respondent asserts that "the criminal justice system of the country is messed up. When we cannot differentiate between the criminals and the justice, the judge can be bribed and allow a criminal to go scot-free, with no trace of the crime or criminal". Crime in the Borokiri area (and this may apply to some other neighborhoods in the global south) is affected by the gross misconduct and the institutional failure of the criminal justice system. Crimes are committed with no trace of evidence because those who commit these crimes are covered up by the same criminal justice system that should fight against crime and protect the rights of residents. This assertion is contrary to the assumption of the routine activity theory that the presence of capable guardians shields against crime; in this situation, it exacerbates crime. Consequently, crimes including kidnapping, robbery, drug trafficking, and rape continue to rise because the system has not used its full weight to clamp down on it.

Furthermore, in agreement with [31], this research argues that in sections of the cities where the fabric of social life has been hampered by economic depression such as unemployment as in the case of Borokiri in Port-Harcourt, the city managers are faced

with challenges in the provision of essential services such as decent housing, educational facilities, and quality health care for their residents. Moreover, this research argues that crime vulnerability, especially in the Borokiri context, is correlated with services such as the aforementioned. In agreement with the social disorganization theory, neighborhoods and societies facing a challenge in the provision of decent housing and quality educational facilities are at risk of crime. The situation is exacerbated in the global south as housing is not seen as a social service—even those which are seen as social services, including water and education, are not efficiently provided, and residents now live in dehumanizing houses which have made the situation more complex to manage. This study shows that the main factors in crime in urban centres are homelessness and deprivation, poverty and unemployment, youth marginalization, the disintegration of culture, lack of urban planning, and neglect of public space. This in turn makes room for delinquency and crime, which gradually becomes a sub-culture. According to a clan head (ID-00018) in the Borokiri area, "crime occurs regularly in this neighborhood occupied by low-income earners. The youth and young people are the ones involved mostly in these crimes. They are mostly involved in the robbery, and petty theft and unemployment are the top factors of these crimes, I would say the neighborhood is security-threatened".

Several arguments abound as to whether social disadvantage is the main cause of crime. While research argues that social disadvantages or economic deprivation are not the causes of crime but critical factors, and that heterogeneity is correlated with crime propensity, this research argues that social disadvantage and economic deprivation/marginalization are the root causes of crime, especially in the Borokiri context, and that heterogeneity has no relationship with crime because both natives and non-natives commit crime. Being an indigene or a non-indigene has nothing to do with the propensity to commit a crime, since as far as Borokiri is concerned, both commit crimes. According to a respondent, a resident (ID-00035), "crime in this neighborhood is something anybody can associate with, so heterogeneity has nothing to do with crime, both indigenes and non-indigenes can commit a crime. Not being satisfied with daily earnings can lead to crime. Committing of crime is a personal decision where language and ethnicity cannot be a barrier". Another participant, a resident (ID-00039) stated that "in my opinion, heterogeneity doesn't relate to crime, anyone and everyone are capable of committing a crime, and indigenes and non-indigenes commit a crime—no one is an exception, same way neither being rich, or poor relates to crime: even language and ethnicity doesn't relate to crime".

Furthermore, this research suggests that residents who are not comfortable with the lifestyle set up in the neighborhood withdraw from community matters as they feel that this is an avenue via which they might be victimized. This has led to population turnover in the form of residential mobility, which weakens communication, hinders communal goal actualization, and slows down overall development. In the Borokiri area, several residents have withdrawn and relocated from their former places of residence to other places where they can live in peace. However, some residents do not feel or see the necessity for mobility as they are becoming acclimatized to the turns of events, and this is very worrisome with an increased risk of higher offenders, contrary to the assertion of the social disorganization theory, which holds the assumption that residential flight is as a result of a high level of vulnerability. In the words of a respondent, resident (ID-00032): "I have lived here for close to fifteen years. I would say the crime rate now is increasing because of the hardship, but I have learnt to live with the events", and from resident (ID-00036): "I have been living in this area for a long time now with most of my neighbors. Nothing the eye has not seen to cry blood; we are used to the situation it is just for you to always be on guard".

Drawing from the discussion above, we can see clearly how these intrinsic factors concerned with demographic and socio-economic characteristics play significant roles in influencing the probabilities of crime and the vulnerability of cities in the global south. From this research, the issue of population turnover is not a significant factor in the rise in crime. As shown in the research, many residents feel very comfortable within the neighborhood. Many have stayed over a long period and do not see a reason for relocating.

From their response, population turnover is not significant, as residents have now adapted to the situation. It is more a matter of caution than fear of crime because crime is ever-present. Moreover, the existence of a link between crime and unemployment is ambiguous, both in terms of its nature and the strength of the association. The exclusion of a significant variable like unemployment in the sociological analysis of crime is a huge omission. It is widely assumed in research that most criminals are unemployed, despite various studies showing otherwise. However, most people who choose to engage in illegal activities are employed. As a result, we concentrate on these aspects to provide a more comprehensive picture of the relationship between crime and unemployment.

One of the critical causes of crime is income disparity. As previously stated, criminal acts are driven by economic motivators. Wage disparities have widened significantly in the last 30 years, and crime rates have risen dramatically at the same time. This research suggests that there is a link between crime and inequality, but care must be taken to separate inequality from poverty's effects. Growth in inequality has a criminal-inducing effect by lowering an individual's moral threshold, a phenomenon known as the envy effect. As a result, rising inequality will positively affect (at least some) people's propensity to commit a crime. The next natural step is to figure out what produces inequality in the first place. Young men are more likely to engage in criminal activities than the general population. There is an increase in participation of disadvantaged young men, particularly the less educated, and more recently the educated too, due to the rise in unemployment. Crime has long-term adverse effects on the opportunity to be employed and committing a crime is only economically rational in the short term.

## 5. Recommendations

Further emphasis should be placed on the significance of education in the fight against crime. The level of education gained, and an individual's economic and social background appear to be inextricably linked to crime. The idea that criminals are less educated and are from lower socioeconomic origins than non-criminals no longer hold. With a high number of graduates, the unemployment rate is running extremely highly. As a result, low education as a factor in criminal behaviour is no longer valid. This idea will enable us to redirect our focus to develop effective and appropriate policies to sustain education, improve its quality and create employment opportunities. This factor is intertwined with other factors that influence and decide crime rates, such as age, unemployment, and inequity.

The majority of the contributions on the effects of education on crime emphasize how education improves people's skills and talents, increasing the number of people who return to a legitimate job and boosting the opportunity costs of illicit behaviour, but there is a failure to emphasize the quality of education in terms of staffing, facilities, and a decent learning environment to produce school leavers who are employable and can take up employment opportunities. However, there are benefits to education that are not considered by individuals, implying that the societal return on education is greater than the private return. Individual preferences are influenced by the indirect effects of education.

The assumption that education reduces crime is fast becoming ambiguous, as being educated may have significant social benefits that individuals do not consider and have gradually been downplayed, so educational centres are gradually becoming breeding grounds for multiple forms of crime such as cultism, vandalism, drug abuse, prostitution, political uproar, and most recently cybercrime. The result of this research has explicitly established that crime is highly correlated with a weak educational system, particularly in the global south. To develop effective approaches for reducing crime, the reward from education (employment opportunities) must be explicit; if not, it may be counterproductive. Given the high social costs of crime, even slight reductions in crime connected with improved opportunities after education could be economically significant.

Furthermore, this research investigated the influence of socio-economic, demographic, and physical characteristics on crime and how they impact crime. However, there is a need

to specifically investigate the influence of the condition of the urban fabric; investigation on this aspect wasn't achieved, which is one of the limitations of this study, and more research is needed to scrutinize the broken windows theory further, involving a potential longitudinal study of crime in a given place.

## 6. Conclusions

Despite the limited sample size, and the narrow scope adopted in this research which focused on lived experiences and concerns socio-economic and demographic characteristics, the study gives an insight into the significance of socioeconomic and demographic characteristics that are exacerbating the level of criminal activities and behaviour. Economic deprivation can lead to stress, strain, family maladjustment, neglect of the importance of education, social vices and association with bad company [40]. All of the above-mentioned factors could lead to delinquency, particularly among youth. These factors are highlighted by the responses from the lived experiences of the respondents in the neighborhood. These factors that exacerbate crime show that residents may easily join anti-social groups whose agenda is only to make society restive and fragile and create an atmosphere of social tension. The respondents are residents of the community who have lived within the area for quite a long time and have either been offenders or victims and so clearly understand the situation.

As crime continues, it raises questions among the residents: what is happening? What factors cause this? Where do we go from here? How can we manage this? There have been constant debates in the social sciences about issues of economic deprivation and how they affect crime, but these have failed to look at the fact that the social structure of the society is experiencing a shift from the norm (social changes). This change is exacerbated by unemployment, economic deprivation, inequality and changes in the concept of the value of education. In other to tackle the challenges of crime in our society, the impact of unemployment, economic deprivation, and inequality must earnestly be placed as a top priority in the goal of national development.

This study can enable the government, city and urban managers, and policymakers to evaluate the current economic situation of Nigerians. Nigeria's poverty rate is about 71%, using the world bank income poverty threshold of $3.20 per day, which means that over a 150 million Nigerians are poor as of the year 2021 [67]. There is an imperative need to develop a well thought out and thorough strategy to alleviate poverty, especially in urban centres. Since unemployment, economic deprivation, and inequality are the main challenges, there should be proper investment and industrial development schemes, informal sector activities and skill acquisition for the youth. The local authorities still have a role to play by running a more inclusive and participatory process of development and decision making, also developing more social activities to shield young people from the influence of bad company.

**Author Contributions:** Conceptualization, C.C.O. and A.H.K.; Methodology, C.C.O.; Software, C.C.O.; Validation, A.H.K. and C.C.O.; Formal Analysis, C.C.O.; Writing the original draft, C.C.O.; Visualization, A.H.K.; Supervision, A.H.K.; Project administration, C.C.O.; Funding Acquisition, C.C.O. All authors have read and agreed to the published version of the manuscript.

**Funding:** The research received no external funding.

**Institutional Review Board Statement:** The Study was conducted in accordance with the Declaration of Helsinki and approved by the University of the Witwatersrand Huan Ethics Research Committee (Non-Medical) with protocol number H20/04/18.

**Informed Consent Statement:** Informed consent was obtained from all subjects involved in the study.

**Acknowledgments:** The authors wish to acknowledge the efforts of Joy Ukamaka Ogbazi of the University of Nigeria, Hassan Elmouelhi of the Technical University of Berlin, Jackson Sebola-Samanyanga of the University of Johannesburg, and Tespang Leuta of the University of the Witwatersrand for their constructive criticism of this documents at the proposal stage. We also wish to thank

the Planning and Housing Graduate Research Committee of the School of Architecture and Planning, University of the Witwatersrand, for a grant for carrying out this research.

**Conflicts of Interest:** The authors declare no conflict of interest.

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
