# Peer review of "An Exploratory Study of Crime: Examining Lived Experiences of Crime through Socioeconomic, Demographic, and Physical Characteristics"

_urbansci, doi:10.3390/urbansci6030043_

Round 1
Reviewer 1 Report
General observation:
- The structure of the paper should clearly highlight Introduction, Literature review, Methodology and data, Results and discussion and Conclusion (or Introduction, Materials and Methods, Results, Discussion, Conclusions, like in template – but I recommend first option). The document must comply with this structure.
- Text quotes (written in italic in “quotation marks” – even are ok like ideeas) and other elements (like text in multiple sections) don’t address the rules of the journal. Should be aligned with the requirement of the journal’s template
- Citing figures and tables don’t address the rules of the journal and some of them are missing. Should be completed and aligned with the requirement of the journal’s template
- It is an eminently descriptive article - I did not identify the personal contribution of the authors (not even the novelty elements, unfortunately - except for a few timid, general recommendations) - I think it should be processed a little bit to highlight the contribution authors for a solution in the sense approached by the subject (Eg. "Mathematical" - or other, but scientifically based - correlation between crime, population density, standard of living, employment, demography, level of corruption, education etc.)
- Introduction:
The section of introduction should include: the context of the study, the gap in literature that the present paper intends to cover, which is the used methodology, which are the main results presented in short, which is the originality of this paper, the main implication policy of these results and a short plan of the rest of the paper. Some of them are missing. I recommend to fill them (e.g. the plan of the rest of the paper should be put at the end of Introduction)
- State of Research in Crime
The chapter is intended to be (probably) perceived as a Literature review, but in this case it should be renamed as such. In addition, the literature research review section should be a critical analysis of the authors on the analyzed topic field and not a series of articles in this field. Actually all the first two chapters should be rewritten and reorganized in order to get a critical analysis if the research field through the lens of the authors.
- Crime in Nigeria
The enumerations and descriptions are positively appreciated, but would be right to emphasize the role of these descriptions and to possibly deliver a comparison that would lead to a clear idea / conclusion.
- Choice of Research Methods and Sampling
- The chapter that is numbered "4.0." it starts well and the text falls into the area (can be renamed Methodology and data or Materials and Methods), but is not completed and is followed by a subchapter (???) numbered "5.1.", which is called "Results" (???)
- The Chapter 5.1. falls into the area Materials (preambul for study)
- The Chapter 5.2. (Significant Induction for Theory Modification and Development.) falls into the area of Discussions and Recommendations (because are some timid recommendations there), but should be segregated much more clearly into Discussions (at the beginning of the chapter) and Recommendations (at the end of the chapter) much consistent. The text completed by the authors could be take into consideretions, but I recommend to be rearranged
- Conclusion
The information is ok.
- References:
The references don’t address the rules of the journal. Should be aligned with the requirement of the journal template
Author Response
Dear Reviewer,
We have addressed all issues raised on a point by point basis, I have attached a copy of the reviewed work.
Thanks

Reviewer 2 Report
Introduction
The research question and subsequently objectives should be stated
Methodology
The data collection should be better explained. How data were obtained? What instrument for data collection has been used and which is its main structure?
Results
The results have to highlight the main findings of the research. The respondents' assertions should be used only to support these findings. They should not take up the largest space of this section.
The results section should be at the same level with the rest of sections. It cannot be labelled as 5.1 as the section 5 do not exist.
It is not very clear what is the content of the subsection 5.2. It is a discussion section or it belongs to the conclusion section?
Conclusion
In this section should be highlighted the main research findings and their relationship with other studies from literature. The implications for academia and business should be stated and most thoroughly addressed. The limitations and further research directions should be stated.
Author Response
Dear Reviewer,
We have attended to all questions/corrections raised by you in the updated manuscript.
Kindly see attached.

Round 2
Reviewer 1 Report
The section of introduction should include: the context of the study, the gap in literature that the present paper intends to cover, which is the used methodology, which are the main results presented in short, which is the originality of this paper, the main implication policy of these results and a short plan of the rest of the paper. Some of them are missing. I recommend to fill them (e.g. the plan of the rest of the paper should be put at the end of Introduction)
2. Literature review
- The information is quite ok, even the section is not a critical review of the authors on the analysed topic.
- First part of the chapter’s text don’t address the rules of the journal regarding the citation mode – should be aligned with the requirement of the journal’s template
3. Methodology and data - ok
4. Results and discussionS (not discussion) – quite ok
- Because there are no Mathematical Components (Theoremes, Axioms, etc.), text quotes (written in “quotation marks” – even are ok like ideeas) don’t address the rules of the journal - should be aligned with the requirement of the journal’s template. In this way, I recommend to be listed (written in order) in Appendix A (see the rules of the journal) - practically the respective texts to be grouped (as they are now) in an annex before References.
- The chapter that is numbered "4.2. - Recommendations" it starts well, but chapter "4.1." is missing – I think that Recommendations should be chapter no. 5
5. Conclusion – ok, but should be numbered like „6”
6. Acknowledgments (not AcknowledgEments - see the rules of the journal) and Conflicts of Interest (not Declaration of interest statement for research - see the rules of the journal) - ok, but should be numbered both like „7”
7. References
Not all the references address the rules of the journal - should be aligned with the requirement of the journal template
Author Response
The Editor
Urban Science Journal,
MDPI Urban Science Editorial Office
St. Alban-Anlage 66, 4052 Basel, Switzerland
E-Mail: [email protected]
https://www.mdpi.com/journal/urbansci
Review of our Journal paper entitled:
An Exploratory Study on Crime: Examining Lived Experiences on Crime Through Socioeconomic, Demographic, and Physical Characteristics.
Dear Editor, we thank you and the reviewers for taking the time to read our manuscript and provide helpful feedback. Your helpful and insightful feedback influenced the current version's potential enhancements. The authors have carefully reviewed the comments and attempted to respond to each one. We trust that after rigorous modifications, the manuscript will match your high requirements. If there are any further constructive remarks, the authors would appreciate hearing from you. The point-by-point responses are listed below (Reviewers 1 ). In the manuscript, all changes have been made.
Sincerely,
RESPONSE TO FIRST REVIEWER COMMENT ON A POINT-BY-POINT BASIS
REVIEW REPORT ROUND 2
|
Comments from first Reviewer |
Response |
1. |
Introduction:
The section of introduction should include: the context of the study, the gap in literature that the present intends to cover, which is the used methodology, which are the main presented in short, which is the originality of this paper. Some of them are missing. I recommend to fill them (e.g., the plan of the rest of the paper should be put at the end of introduction). |
Corrections effected. Refer to paragraphs 5 – 7 of the section of the introduction. |
2. |
Literature review:
- The information is quite ok, even the section is not a critical review of the authors on the analysed topic. - First part of the chapter’s text don’t address the rules of the journal regarding the citation mode – should be aligned with the requirements of the journal’s template. |
Citation aligned to address the rules of the journal. Refer to first paragraph of the literature review section. |
3. |
Methodology and data:
Ok. |
|
4. |
Results and discussions:
Results and discussionS (not discussion) – quite ok - Because there are no Mathematical Components (Theoremes, Axioms, etc.), text quotes (written in “quotation mark” – even are ok like ideas) don’t address the rules of the journal – should be aligned with the requirements of the journal’s template. In this way, I recommend to be listed (written in order in Appendix A (see the rules of the journal) – practically the respective texts to be grouped (as they are now) in an annex before references. |
|
5. |
Recommendations:
- The chapter that is numbered “4.2. – Recommendations” it starts well, but chapter “4.1.” is missing – I think that Recommendations should be chapter no. 5 |
Corrections effected. See the section of the Recommendations. |
6. |
Conclusion:
Conclusion – ok, but should be numbered like “6” |
Corrections effected. See the section of the Conclusion. |
7. |
Acknowledgments:
Acknowledgments (not AcknowledgEments – see the rules of the journal) and Conflicts of Interest (not Declaration of interest statement for research – see the rules of the journal) – ok, but should be numbered like “7” |
Corrections effected. Refer to the section of the Acknowledgment and the Declaration of interest statement. |
8. |
References:
Not all the references address the rules of the journal – should be aligned with the requirements of the journal’s template. |
Corrections effected. All references have been aligned to address the rules of the journal’s template. |

Reviewer 2 Report
I recommend to authors to give answers point by point to my requests. It is not usual to respond to reviewers with the text of entire article from which every body can understand what he/she wants.
Author Response
The Editor
Urban Science Journal,
MDPI Urban Science Editorial Office
St. Alban-Anlage 66, 4052 Basel, Switzerland
E-Mail: [email protected]
https://www.mdpi.com/journal/urbansci
Review of our Journal paper Title
An Exploratory Study on Crime: Examining Lived Experiences on Crime Through Socioeconomic, Demographic, and Physical Characteristics.
Dear Editor, we thank you and the reviewers for taking the time to read our manuscript and provide helpful feedback. Your helpful and insightful feedback influenced the current version's potential enhancements. The authors have carefully reviewed the comments and attempted to respond to each one. We trust that after rigorous modifications, the manuscript will match your high requirements. If there are any further constructive remarks, the authors would appreciate hearing from you. The point-by-point responses are listed below(Reviewers 2). In the manuscript, all changes have been effected.
Sincerely,
RESPONSE TO SECOND REVIEWER COMMENT ON A POINT-BY-POINT BASIS
|
Comments from 2nd Reviewer |
Response |
1. |
Introduction The research question and subsequently objectives should be stated. |
Correction effected, see page 3 paragraph 1. |
2. |
Methodology The data collection should be better explained. How were data obtained? What instrument for data collection has been used and which is its main structure? |
Corrections effected, see Page 3 paragraph 2.
3.0 Methodology and data Paragraph 1, Line 6 and 7 Paragraph 2.
|
3. |
Results The results have to highlight the main findings of the research. The respondents' assertions should be used only to support these findings. They should not take up the largest space of this section. The results section should be at the same level with the rest of sections. It cannot be labelled as 5.1 as the section 5 do not exist. It is not very clear what is the content of the subsection 5.2. It is a discussion section or it belongs to the conclusion section? |
Corrections effected, see 4.0 Results and Discussion.
Correction effected.
Corrections Effected. |
4. |
Conclusion In this section should be highlighted the main research findings and their relationship with other studies from literature. The implications for academia and business should be stated and most thoroughly addressed. The limitations and further research directions should be stated. |
Corrections effected |

Round 3
Reviewer 2 Report
The authors made the suggested improvements.